# The Behavior of Long Thin Rectangular Plates under Normal Pressure—A Thorough Investigation

**DOI:** 10.3390/ma17122902

**Published:** 2024-06-13

**Authors:** Gilad Hakim, Haim Abramovich

**Affiliations:** Aerospace Structures Lab, Faculty of Aerospace Engineering, Technion—Israel Institute of Technology, I.I.T., Haifa 32000, Israel; ghakim@outlook.com

**Keywords:** square thin plate, aspect ratio, small linear deflection, large deflection, membrane stress, finite element analysis, movable and immovable edges

## Abstract

Thin rectangular plates are considered basic structures in various sectors like aerospace, civil, and mechanical engineering. Moreover, isotropic and laminated composite plates subjected to transverse normal loading and undergoing small and large deflections have been extensively studied and published in the literature. Yet, it seems that the particular case of long thin plates having a high aspect ratio appears to be almost ignored by various scholars despite its engineering importance. The present study tries to fill this gap, yielding novel findings regarding the structural behavior of long thin plates in the small- and large-deflection regimes. In contrast to what is normally assumed in the literature, namely that a long plate with a high aspect ratio can be considered an infinitely long plate, the present results clearly show that the structural effects of the ends continue to exist near the remote ends of the long plate. An innovative finding is that long plates would (only on movable boundary conditions for the large-deflection regime) exhibit a larger mid-width displacement in comparison with deflections of infinitely long plates. This innovative higher deflection appears for both small and large-deflection regimes for both all-around simply supported and all-around clamped boundary conditions. This new finding was shown to be valid for both isotropic and orthotropic materials and presents a novel engineering approach for the old assumption well quoted in the literature that a relatively long plate on any boundary condition can be considered an infinite plate. Based on the present research, it is recommended that this assumption should be used carefully as the largest plate mid-deflection might occur at finite aspect ratios.

## 1. Introduction

Thin and thick plates are known to be the basic blocks in various engineering sectors, like aerospace, civil, and mechanical engineering. Moreover, these plates may undergo large deflections, and for some applications, their length-to-width ratio or aspect ratio (*AR*) tends to be large.

As a result, many research articles and books have been published on thin rectangular plates with transverse loads, for both small and large deflections. Many of them include (in various forms) a general statement that “long plates with a high *AR* can be treated as an infinitely long plate”; i.e., the remote ends do not affect the plate’s mechanical performance and the plate would experience its highest out-of-plane deflection. For instance, Timoshenko [1] (p. 422) states that “*the deflections of finite plates with b/a < ⅔* (where *b* and *a* are the width and length of the plate, respectively) *are very close to those obtained for an infinitely long plate*”. Note that this statement would relate to *AR* > 1.5. Also, the stresses for *AR =* 2 are said to be 10% lower than those for an infinitely long plate. Reddy [2] (p. 246) provides information for the midpoint deflection and bending moments for plates with *AR =* 2 and less. On p. 249 of Reddy’s work [2], information is given for *AR =* 3 and less for the bending moment load, while on p. 314, a buckling case is considered for *AR =* 6 and less. Bakker et al. [3], in their article, refer to an *AR* of up to 2. Longer plates are not considered. Further information regarding the behavior of plates under loading can be found in [4,5,6,7,8,9]. Wang and El-sheikh [10] present deflection information up to *AR =* 10, restricting themselves to immovable boundary conditions (namely, the plate edges cannot move relative to their support frame). Shao [11], in his thesis, investigated high-*AR* plates up to *AR =* 5, with movable edges, but the edges were forced to remain straight. The reason for this *AR =* 5 limit is that (p. 2) “*beyond this value, the behavior of the plate is nearly as a strip*”. Razdolsky [12] also investigated thin rectangular plates up to *AR =* 4, with simply supported immovable edges. He presents deflections and stresses in a graphical form for specific points on the plate. For the general case, a rather complex algorithm is suggested using multiple summations of trigonometric functions. Ostiguy and Evan-lwanowski [13] also checked the influence of the aspect ratio on the dynamic response and stability of rectangular plates under the large-deflection regime. They found that “*AR* plays a crucial role in determining the stability of rectangular plates”.

The present study presents a thorough investigation on the behavior of long plates under normal pressure, challenging the above-quoted statement by checking the performance of high-*AR* plates, for both deflections and stresses. Innovative, interesting results suggest another perspective for these long plates, showing higher deflections at specific mid-width points.

## 2. Materials, Methods, and Results

To investigate the behavior of long plates with small deflections, the classical Navier solution for the midpoint small deflection of a thin rectangular plate was analyzed for large *AR*s. The solution presented in Reddy’s work [2] on p. 230 is brought here again for clarification and is presented in Equation (1).
(1)$$w_{\max}=\frac{16qb^{4}}{D\pi^{6}}\overset{\infty }{\underset{n=1,3,5\ldots }{\sum }}\overset{\infty }{\underset{m=1,3,5\ldots }{\sum }}\frac{{\left(-1\right)}^{\frac{m+n}{2}-1}}{mn{\left(m^{2}{\left(\frac{b}{a}\right)}^{2}+n^{2}\right)}^{2}}$$

Then, the midpoint deflection expression was calculated for several *ARs* ab using the polycarbonate data presented in Table 1. The *E* and *ν* values of this isotropic linear elastic material were taken from manufacturer’s commercial data sheets [14,15].

The calculated midpoint deflection for a square plate was found to be *w*_max_ = 13.9 mm.

The relative mid-deflection, defined as the ratio between the actual lateral displacement and the maximal deflection of a square plate, as a function of *AR* is depicted in Figure 1.

From the graphs presented in Figure 1, it is obvious that plates with a high finite *AR* (in the region of *AR* = 10) would deflect more than an infinitely long plate. The difference is just 1%, and can be considered negligible, but it still contradicts the well-known general statement that “long plates with a high aspect ratio *AR*, can be treated as an infinitely long plate”.

To check the importance of the boundary conditions, an all-around clamped plate was investigated with increasing aspect ratios, in a similar way to that shown above for all-around simply supported boundary conditions. Since a simple solution for the case of all-around clamped boundary conditions is not available, a finite element analysis (FEA) method was used. The analysis code was Simcenter Femap with Nastran ver. 2021.1 from Siemens [16]. More details about the FEA are given in Appendix A.

The dimensions and the mechanical properties, presented in Table 1, were used to yield the two graphs presented in Figure 2. A pressure load of 25 Pa was used to have a small midpoint deflection of about half-plate thickness, while the midpoint deflection for a square plate was found to be *w*_max_ = 1.1035 mm.

As for the all-around simply supported case, as well as for the all-around clamped case, there is a small peak at *AR* = 3, which is about 0.5% larger than the infinite-length deflection case. So again, although the difference is just 0.5% and can be considered negligible, it still contradicts the well-known general statement that “long plates with a high aspect ratio, *AR*, can be treated as an infinitely long plate”.

The present study will now proceed to investigate the large-deflection regime. While, for a small-deflection regime, the in-plane effects are generally negligible, for a large-deflection regime that exceeds the plate thickness, the in-plane effects play a major role and must be considered. The additional elastic energy required to strain the plate in its plane causes the plate to deflect less than expected when using the linear small-deflection theory.

To investigate the behavior of a thin plate with a high *AR*, in the large-deflection regime, rectangular plates were addressed. The geometry and the material data are presented in Table 1. Note that the length is defined in the *x* direction, while the width is in the *y* direction, as shown in Figure 3. To enable large deflections, the plate was loaded with a uniform pressure, *q* = 1000 Pa.

In the plate’s large-deflection regime, the in-plane boundary conditions (BCs) have a significant influence on both the deflection and the generated membrane stresses. Since there are many possible BC combinations, in the present study, we dealt with only two representative BCs: a four-side simply supported designated SSSS where there are no bending moments at the various edges, and a four-side clamped designated CCCC where no rotations are allowed at the various edges.

For the in-plane BC, we used two types, which are described hereafter. The movable four sides, where the plate edges can freely move relative to the support frame in both the parallel and perpendicular directions, are designated as M. The immovable four sides, where the plate edges cannot move relative to the support frame, are written as I.

Finite element analyses (FEAs) for several types of rectangular plates were performed. As for the case of small deflections presented above, the analysis code was Femap 2021.1 from Siemens, with Simcenter Nastran [16] as the code processor (see Appendix A for details). A nonlinear static analysis was performed for quad-type plate elements with a 50 mm element size. The nonlinear code increased the load in 20 steps, while in each step, the deflections and stresses were recalculated and used as a starting point for the next step. Each of these steps had internal iterations to verify its convergence. Upon completion of the run, all final deflections and membrane forces of the entire plate were transferred to an Excel (v2405) sheet. Also, graphic pictures of the FEA results were saved for further processing. Then, the data and figures were further evaluated for significant findings. For further details, see Hakim and Abramovich [17].

One should note that during nonlinear analysis, due to the large-deflection status, the plate elements would change their special direction. Therefore, one must consider the load direction acting on the plate elements. Two types of distributed loads are commonly used within the FE code. The first one is a vector-oriented load direction, in which the load direction is fixed in the space and does not follow the element direction’s special changes. The second one is a pressure load type that operates perpendicularly to the element surface and follows the special change in the element direction, namely, a follower force. These two load types represent clear physical situations. The vector-oriented distributed load represents snow load, where the load direction is the weight of the snow acting downward. The distributed pressure load represents a wind load that operates perpendicularly to the plate element and follows its special directions while loaded. In the present study, we used the vector-oriented distributed load, applying it to obtain the plate’s response to distributed equal load.

Typical results are presented in Figure 4, Figure 5, Figure 6 and Figure 7, presenting the calculated out-of-plane deflections, the membrane *x* forces, the *y* forces, and the shear *xy* forces for the SSSS-M case. The numbers near the plates are their respective *ARs*, while SFSF means that the two remote ends (left and right) are free and the two sides (up and down) are simply supported, representing an infinitely long side-supported plate.

The relatively high *Y* membrane force near the ends of the SFSF is probably related to the sudden discontinuity of the plate combined with the Poisson effect. Nevertheless, as the SFSF represents an infinitely long plate, this end effect is ignored here.

Based on the results presented in Figure 4, the influence of the *AR* on the out-of-plane deflections was investigated.

One may expect that the out-of-plane deflection for plates with a high aspect ratio would asymptotically approach the deflection of an infinitely long plate, with the long plate having the highest deflection.

However, checking the deflection against increasing values of *AR* reveals an interesting phenomenon for the movable BC (M) cases. To highlight it, the midpoint deflections for several plates were normalized by the deflection of a square plate (*AR =* 1). The relative deflection is defined as the ratio of the midpoint deflection to the midpoint deflection of a square plate. The resulting graphs are presented in Figure 8 and Figure 9.

As shown in Figure 8, the plate’s deflections for 8 < *AR* < 20 are higher than that of an infinitely long plate with a maximum *AR =* 12. This unexpected behavior persists with other Young’s moduli and other Poisson’s ratios *ν*, including *ν* = 0.

Identical results were obtained using another piece of FEA software, Ansys 2023/R2, as can be seen in Figure 8.

Ansys is a large scientific analysis code described partially in [18,19]. The section used here was Ansys Workbench with its Static Structural finite element section. In the analysis, the large-deflection option was activated to correctly account for the in-plane strains.

To present the behavior shown in Figure 8 more conveniently, an empirical formula describing the relative midpoint deflection, *RD*, as a function of *AR* using a minimum squares regression was performed. Excel’s Solver add-in was used to minimize the squares sum by optimizing the formula coefficients. This empirical formula for the SSSS-M BC case is presented in Equation (2):(2)$$RD=B+C\cdot \exp\left(D\cdot AR\right)\cdot \sin\left(E\cdot AR+F\right)$$

The optimal coefficients were found to be *B* = 9.3078, *C* = −11.167, *D* = −0.20872, *E* = 0.24266, and *F* = 0.90972, with an excellent correlation coefficient of *R* = 0.99996. The valid range of *AR* in (2) is *AR* ≥ 1, while the *RD* value approaches *B* for high *AR*s. The influence of other material properties on this behavior is still to be checked.

The clamped movable CCCC-M case in Figure 9 shows a similar behavior, but with another high range, 6 < *AR* < 15, and a lower maximum point at *AR =* 8.

To demonstrate that the above phenomenon also occurs in other types of plates made of non-isotropic materials, 16 mm multiwall plates were analyzed. Multiwall plates consist of parallel thin-walled sheets connected by vertical ribs, extruded from various plastics to yield a light and yet rigid structure, as seen in Figure 10.

A detailed description of these plates and their mechanical behavior under uniform pressure is presented in Hakim and Abramovich’s work [17].

To correctly account for the complex internal structure, a homogenization procedure was applied, in which 10 independent elastic constants represented the global plate’s elastic response as a solid plate, ignoring the small internal structure features. In Table 2, these constants are listed (see [20] for the homogenization process of the plate).

The above extruded plate, supported with the SSSS-M BC, was analyzed using the Femap code. The chart in Figure 11 shows the same effect as that discussed above.

One should note that this behavior occurs only in plates with a movable BC (M), while plates with an immovable BC (I) do not show this phenomenon, as is presented in Figure 12 for isotropic plates with properties and loading identical to the ones used for movable SSSS boundary conditions.

While trying to identify the origin of this behavior, the width midpoint deflections were checked along several long plates. The plates were identical but with different lengths. Figure 13 shows the mid-width deflection along the plates.

From these graphs, it is obvious that the high-deflection red points are located about six widths away from the plate ends. This might explain the maximum seen in Figure 8 at *AR =* 12, where the two maxima meet each other.

To further investigate the topic in the large-deflection regime and the influence of the type of loading, namely the vector and pressure types described above, a comparison was made between the two cases. Figure 14 demonstrates that both load types present the same behavior, with the pressure load type presenting a smaller deflection.

To understand the difference in the lateral deflections when applying the two types of loading, as presented in Figure 14, the sum of the vertical (in the direction of the *z* coordinate—see Figure 3) nodal reaction forces at the plate edges was calculated. The plate dimensions were 1 m × 12 m and the load value was 1000 Pa (one time with a fixed-direction vector load and one time with a pressure follower load). The reaction with the fixed vector load was exactly 12 kN (load multiplied by the area), while the reaction with the pressure follower load was only 10.7 kN. The reduced reaction in the second case was caused by the change in the element direction with the pressure force that followed the change. This reduction induced the smaller deflections seen in Figure 14, as the vertical force was reduced.

## 3. Discussion

The classical question regarding thin rectangular plates, namely, “*Above which aspect ratio (AR) would the plate behave like an infinitely long plate?*”, cannot be answered intuitively. The present study clearly demonstrates that the influence of the plate’s ends on the deflection and on the in-plane stresses penetrates from the ends into the plate’s midpoint differently. Moreover, when the plate becomes longer, the end effects do not disappear but, rather, remain stuck to the moving-away ends.

In addition, for out-of-plane deflections, the end influence penetrates towards the plate’s midpoint from each end by 5–6 times the width.

For tensile membrane *X* stresses (or forces), the influence also penetrates towards the midpoint by five times the width.

For tensile membrane *Y* stresses (or forces), the influence penetrates towards the midpoint by one width.

For shear membrane *XY* stresses (or force), the influence also penetrates towards the midpoint by one width.

Another significant finding stemming from the present study is that the midpoint deflection of a specific *AR* range is higher than that of an infinitely long plate. Higher deflections occur in SSSS-M’s case six widths away from the ends towards the plate’s midpoint. This is evident for the small-deflection linear Navier solution, all-around clamped plate with small-deflection, and large-deflection regimes with movable edges. The authors believe that this is a new finding that has not been reached before in the literature and will encourage future research on it.

This innovative finding has some implications when designing plates for given specified deflections. The common assumption is that infinitely long plates have the highest deflection, more than any rectangular plate with the same width, and therefore they are considered the most conservative design. However, the common assumption described above might be wrong since the present findings prove that a long plate (8 < *AR* < 20 for a simply supported BC and 6 < *AR* < 15 for a clamped BC) in the large-deflection regime would deflect more than an infinite plate, for both simply supported and clamped movable edges. Therefore, caution is suggested when making this common assumption while dealing with long plates under normal lateral pressure.

## 4. Conclusions

Based on the results presented in the present study, the following conclusions can be drawn:High-*AR* rectangular plates (above a certain value) with movable edges cannot be considered infinitely long plates, but rather, the end influences are preserved near the ends.The midpoint deflections of the SSSS-M case with 10 < *AR* < 20 and of the CCCC-M case with 6 < *AR* < 15 are higher than that of an infinitely long plate. A plausible explanation for this phenomenon might be the unusual shape of the deflections for these unique boundary conditions.This behavior has also been detected for small-deflection theory based on the classical Navier solution and all-around clamped plates in the small-deflection regime.The influence of the plate’s ends on the deflection and on the in-plane stresses penetrates from the far ends into the plate’s midpoint in different ways.

## Figures and Tables

**Figure 1 materials-17-02902-f001:**
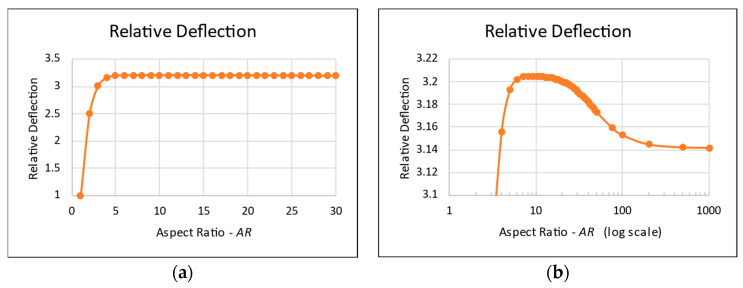
Navier small deflection theory—relative deflection vs. *AR*: (**a**) full scale; (**b**) peak vicinity.

**Figure 2 materials-17-02902-f002:**
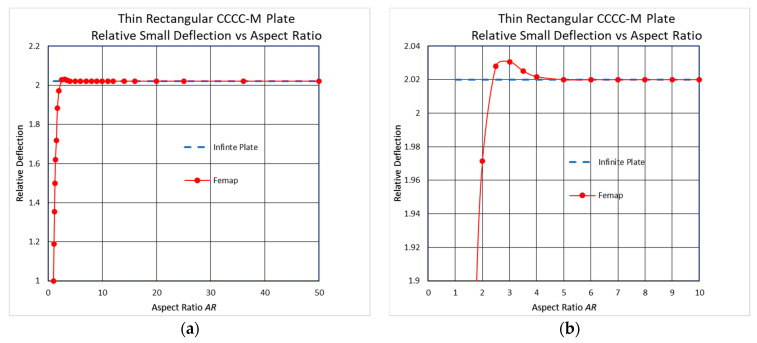
Thin rectangular all-around clamped conditions in the small-deflection regime (FE results)—relative deflection vs. *AR*: (**a**) full scale; (**b**) peak vicinity.

**Figure 3 materials-17-02902-f003:**
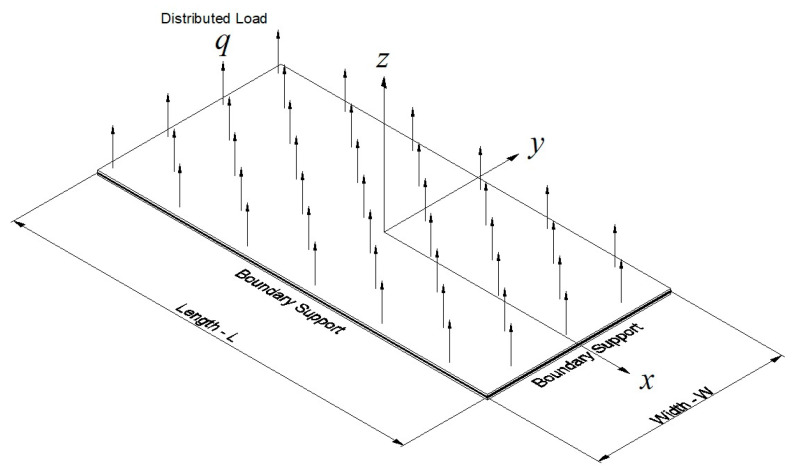
The plate schematic model for the large-deflection regime.

**Figure 4 materials-17-02902-f004:**
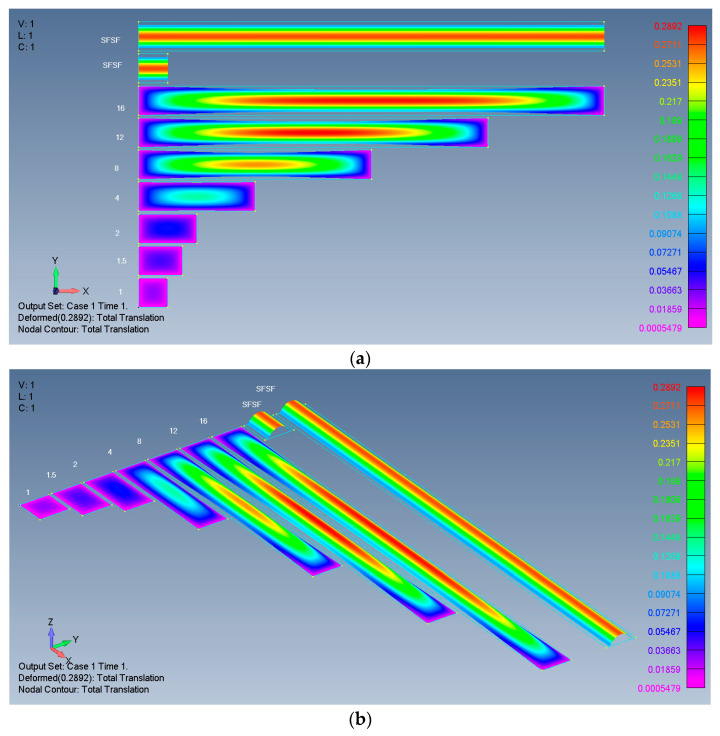
Deflections of rectangular isotropic plates: (**a**) 2D view; (**b**) 3D view.

**Figure 5 materials-17-02902-f005:**
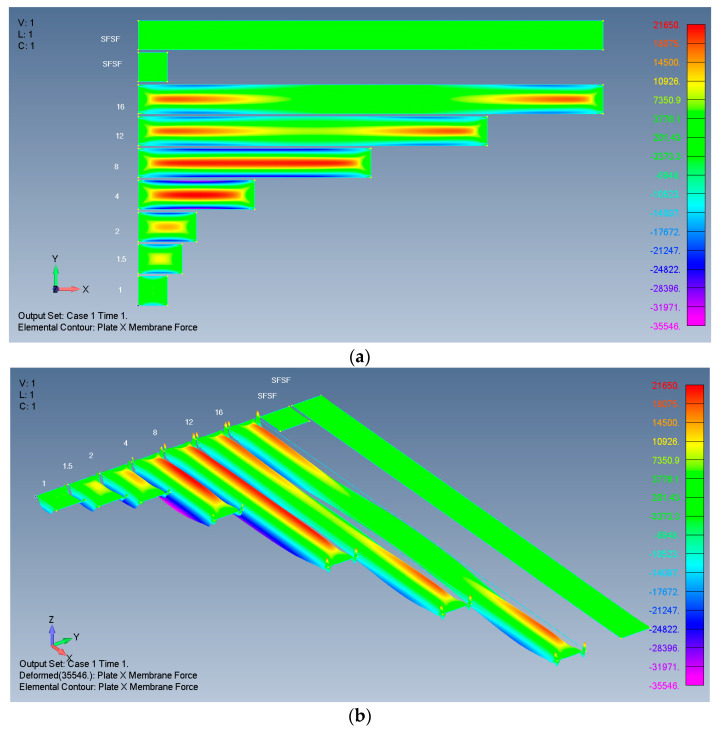
*X* membrane forces of rectangular isotropic plates: (**a**) 2D view; (**b**) 3D view.

**Figure 6 materials-17-02902-f006:**
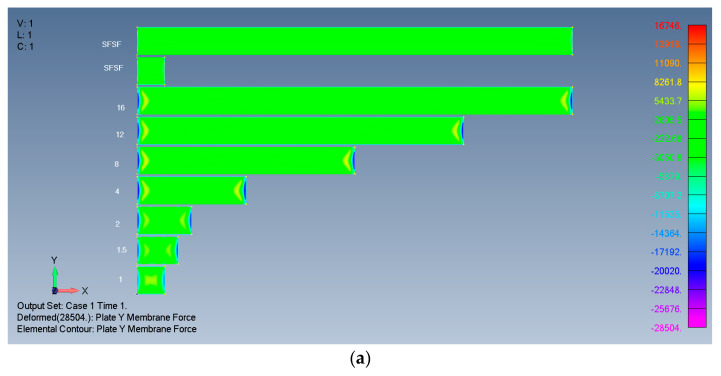
*Y* membrane forces of rectangular isotropic plates: (**a**) 2D view; (**b**) 3D view.

**Figure 7 materials-17-02902-f007:**
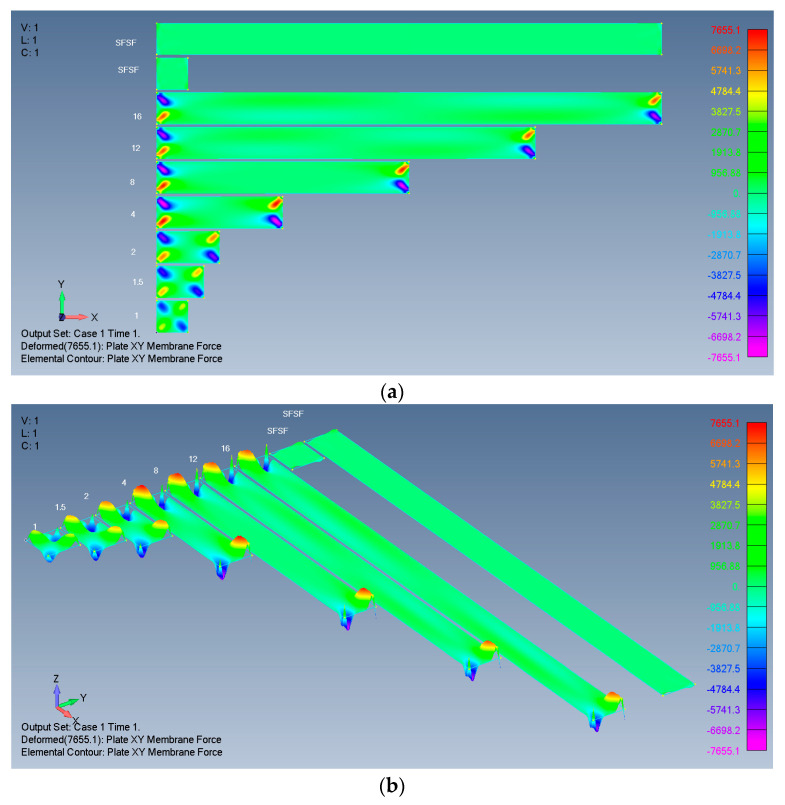
*XY* membrane forces of rectangular isotropic plates: (**a**) 2D view; (**b**) 3D view.

**Figure 8 materials-17-02902-f008:**
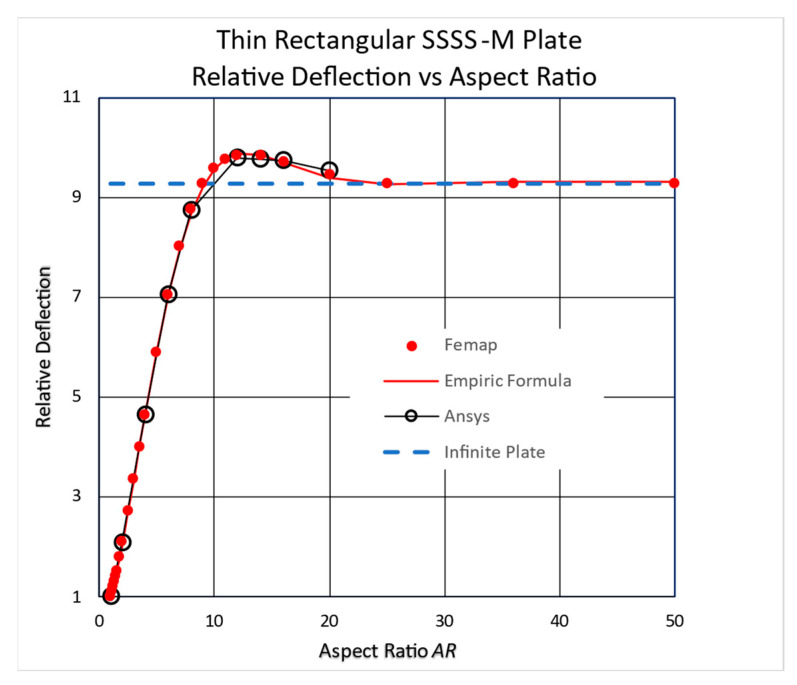
Relative midpoint deflection for high *ARs*—FEAs and empirical formula.

**Figure 9 materials-17-02902-f009:**
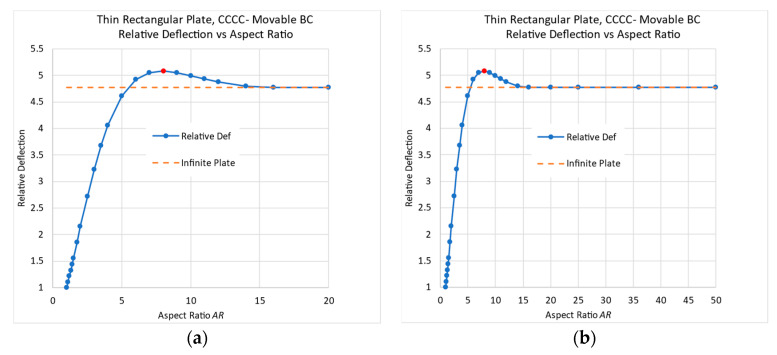
CCCC-M’s relative midpoint deflection for high *AR*: (**a**) up to *AR* = 20; (**b**) up to *AR* = 50.

**Figure 10 materials-17-02902-f010:**
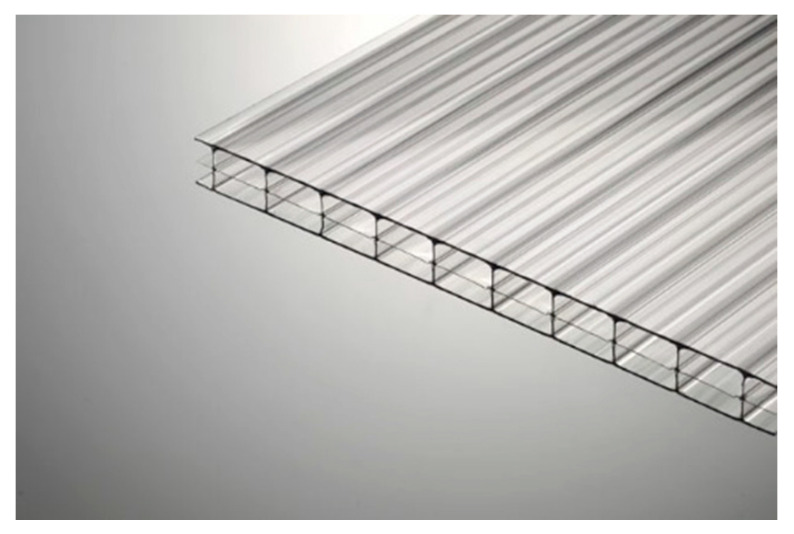
A 16 mm triple-wall plate.

**Figure 11 materials-17-02902-f011:**
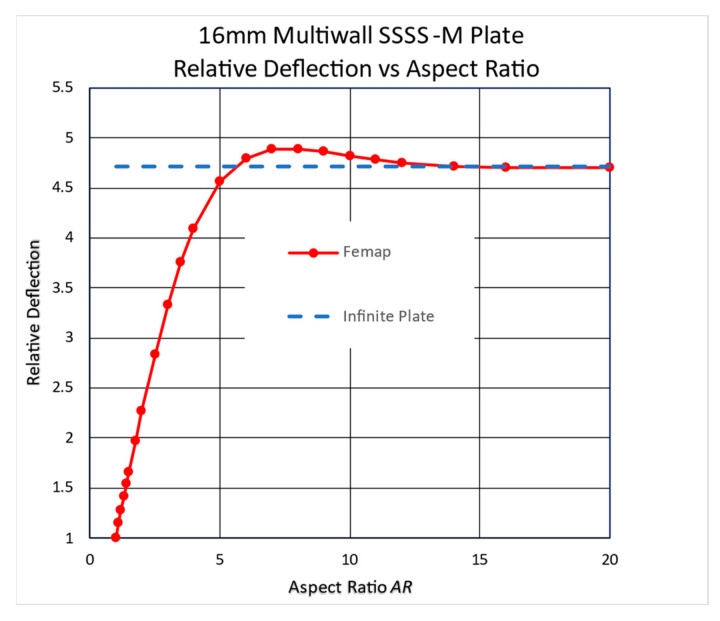
A multiwall plate’s higher deflection.

**Figure 12 materials-17-02902-f012:**
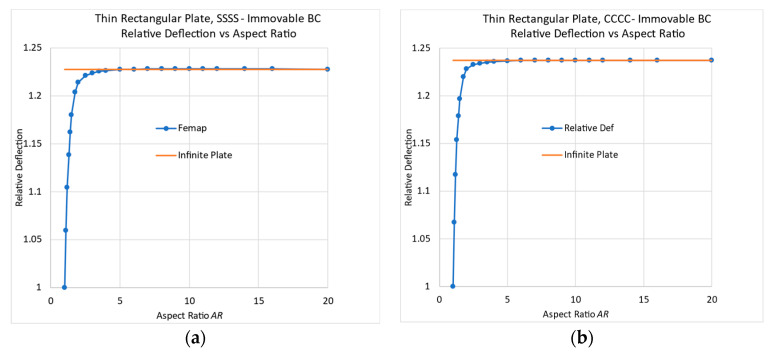
Deflections of rectangular plates on immovable edges: (**a**) simply supported: SSSS-I; (**b**) clamped: CCCC-I.

**Figure 13 materials-17-02902-f013:**
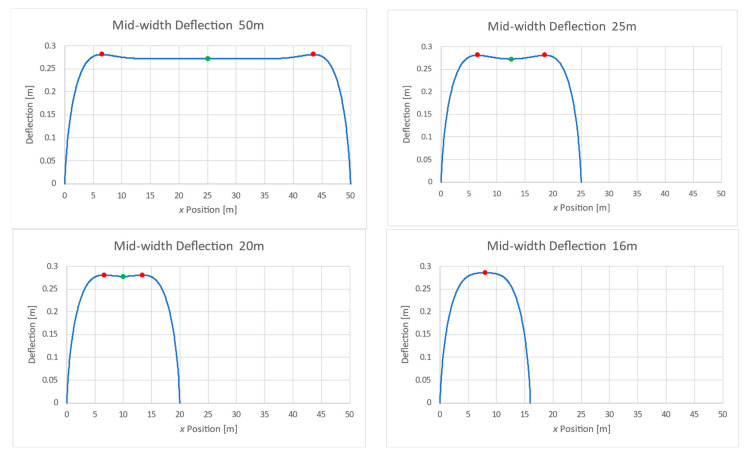
Mid-width deflections of SSSS-M plate with various lengths.

**Figure 14 materials-17-02902-f014:**
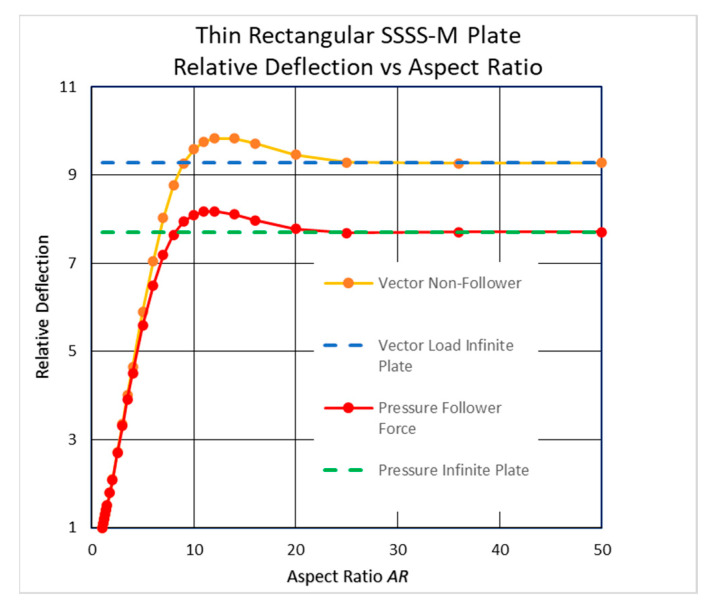
Relative deflection vs. *AR* for a fixed vector load and a follower pressure load.

**Table 1 materials-17-02902-t001:** Structural and material data used to evaluate Equation (1) for various *AR* values.

Young’s modulus *E*	2.4 GPa
Poisson’s ratio *ν*	0.38
Thickness *h*	0.005 m
Length *a*	varies
Width *b*	1 m
Distributed load *q*	100 Pa
The calculated bending rigidity *D D* = *Eh*^3^/12(1 − *ν*^2^)	29.219 Nm
Summation indexes *m* and *n*	1, 3, 5, …, 31

**Table 2 materials-17-02902-t002:** The 16 mm multiwall elastic equivalent constants.

Equivalent *E_y_^t^* modulus	228.56	MPa
Equivalent *E_x_^t^* modulus	318.77	MPa
Equivalent *G_xy_^t^* modulus	83.417	MPa
Poisson’s ratio *ν_xy_^t^*	0.37	
Equivalent *E_x_^b^* modulus	647.69	MPa
Equivalent *E_y_^b^* modulus	370.51	MPa
Equivalent *G_xy_^b^* modulus	13.55	MPa
Poisson’s ratio *ν_xy_^b^*	0.37	
Shear rigidity *S_x_*	136.27	N/mm
Shear rigidity *S_y_*	3.8697	N/mm

The superscript *t* stands for “tension” and *b* stands for “bending”.

## Data Availability

The raw data supporting the conclusions of this article will be made available by the authors on request.

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
