# Peer review of "The Behavior of Long Thin Rectangular Plates under Normal Pressure—A Thorough Investigation"

_materials, 2024, doi:10.3390/ma17122902_

Round 1

Reviewer 1 Report

Comments and Suggestions for Authors

The paper has some interesting points; however, several aspects need improvement and clarification before it can be recommended for publication. Please address the following points thoroughly to enhance the manuscript:

1- The Abstract would benefit from a comprehensive improvement to better highlight the novelty and significance of this work.

2- The novelty of this research could significantly improve with a thorough examination of previous studies and a critical review of literature relevant to the readership of applied thermal engineering.

3- The simulation section, on the other hand, lacks several key aspects, including the engineering and scientific description of the discretization system, convergence tests, assumptions for the model, justification of the chosen system, calculation of equations in terms of the discretized systems, and adequate description of the model. This is essential not only to demonstrate the validity of the numerical studies but also to facilitate data reproducibility. The authors may refer to sources such as (https://doi.org/10.1016/j.applthermaleng.2023.121124) and                           (https://doi.org/10.1016/j.applthermaleng.2017.11.133) for further information.

4- The manuscript, in its current form, has several ambiguities in presentation across various sections, starting from the introduction. To aid the authors in improving the clarity and quality of the paper, please consider addressing the following questions during the revision process:

4-1. How was the finite element analysis methodology applied in this study validated against theoretical and empirical models? Specifically, discuss the choice of Femap with Nastran and Ansys 2023/R2 software, including their non-linear static analysis capabilities and the convergence criteria employed during simulation.

4-2. In what ways do the movable boundary conditions (SSSS-M and CCCC-M) influence the mid-point deflection behavior of long thin plates, and how do these findings challenge the traditional assumption that long plates with high aspect ratios can be approximated as infinitely long plates?

4-3. The study finds that plates with certain high aspect ratios exhibit higher mid-point deflections compared to infinitely long plates. Can you elucidate the underlying mechanical principles that lead to the maximum deflection occurring within specific aspect ratio investigated rages?

4-4. Describe the differences observed between vector-oriented and follower pressure load types on the deflection profiles of long thin plates. How do the changes in nodal reaction forces explain the reduced deflections observed with follower pressure loads compared to fixed direction vector loads?

4-5. Discuss the practical implications of the study's findings on the design and safety considerations of thin rectangular plates in engineering fields such as aerospace, civil, and mechanical engineering. Specifically, how should engineers reconsider the conservative design assumption that infinitely long plates experience the highest deflections under normal pressure?

Author Response

Reviewer # 1: Comments and Suggestions for the Authors

The authors would like to thank the reviewer for his constructive comments.

Our reply to reviewer's comments, are highlighted in yellow

The paper has some interesting points; however, several aspects need improvement and clarification before it can be recommended for publication. Please address the following points thoroughly to enhance the manuscript:

1- The Abstract would benefit from a comprehensive improvement to better highlight the novelty and significance of this work.

The Abstract was modified and improved to highlight the present paper innovation.

2- The novelty of this research could significantly improve with a thorough examination of previous studies and a critical review of literature relevant to the readership of applied thermal engineering.

The novelty of the present study has been discussed in several sections of the paper, with adequate references from previous studies. As the article deals with purely applied mechanical loads, it will be confusing for the reader to add references related to applied thermal engineering. Therefore all the references presented deal with mechanical loads only.

3- The simulation section, on the other hand, lacks several key aspects, including the engineering and scientific description of the discretization system, convergence tests, assumptions for the model, justification of the chosen system, calculation of equations in terms of the discretized systems, and adequate description of the model. This is essential not only to demonstrate the validity of the numerical studies but also to facilitate data reproducibility. The authors may refer to sources such as (https://doi.org/10.1016/j.applthermaleng.2023.121124) and                           (https://doi.org/10.1016/j.applthermaleng.2017.11.133) for further information.

To answer the various topics raised by the reviewer a new Appendix A has been added, in which the whole FEA process is described in detail. As said above, the reviewer’s suggested sources deal with thermal analysis which are not discussed in this paper as it deals purely on mechanical applied loads..

4- The manuscript, in its current form, has several ambiguities in presentation across various sections, starting from the introduction. To aid the authors in improving the clarity and quality of the paper, please consider addressing the following questions during the revision process:

4-1. How was the finite element analysis methodology applied in this study validated against theoretical and empirical models? Specifically, discuss the choice of Femap with Nastran and Ansys 2023/R2 software, including their non-linear static analysis capabilities and the convergence criteria employed during simulation.

The authors could not find in the literature any empirical real test results for high AR plates. Also, theoretical models do not specifically address the present discussed case. Therefore, a FEA was used as a substitute to real tests and reliable theory, without additional validation. The added Appendix A describes the FEA in detail, including the convergence control method. Both Femap and Ansys codes have very good non-linear capability which was required in the present case. The reason to use two different software was to verify that the results were not influenced by an incidental bug in one of the systems and not to compare between the two codes..

4-2. In what ways do the movable boundary conditions (SSSS-M and CCCC-M) influence the mid-point deflection behavior of long thin plates, and how do these findings challenge the traditional assumption that long plates with high aspect ratios can be approximated as infinitely long plates?

The mid-point deflection in CCCC-M is less than SSSS-M since the first BC is more restrictive, resulting in less deflection. As this mid-point deflection is larger than that of infinite length plate, it challenges the common assumption that infinite length plate has the highest deflection.

4-3. The study finds that plates with certain high aspect ratios exhibit higher mid-point deflections compared to infinitely long plates. Can you elucidate the underlying mechanical principles that lead to the maximum deflection occurring within specific aspect ratio investigated rages?

As described in the 2nd conclusion, the reason for this phenomenon may be the interesting shape of the deflection. The authors believe that the published paper will initiates further research of this behaviour.

4-4. Describe the differences observed between vector-oriented and follower pressure load types on the deflection profiles of long thin plates. How do the changes in nodal reaction forces explain the reduced deflections observed with follower pressure loads compared to fixed direction vector loads?

When the applied load is a follower force (pressure), the vertical component of the force becomes smaller since the surface is sloped in high deflection, resulting in less deflection and less vertical boundary reaction in comparison to a fixed direction load.

4-5. Discuss the practical implications of the study's findings on the design and safety considerations of thin rectangular plates in engineering fields such as aerospace, civil, and mechanical engineering. Specifically, how should engineers reconsider the conservative design assumption that infinitely long plates experience the highest deflections under normal pressure?

The practical implications of the present study are discussed and presented in the last paragraph of the Discussion section of the manuscript.

Reviewer 2 Report

Comments and Suggestions for Authors

The paper investigates the structural behavior of long thin rectangular plates under normal pressure, revealing that plates with high aspect ratios exhibit higher mid-width deflections than infinitely long plates, challenging existing assumptions. The study utilizes classical Navier solutions and finite element analysis to demonstrate that the end effects of these plates persist significantly, influencing their overall mechanical performance. Generally speaking, the paper is well written; however, the following are my major concerns:

1. The introduction provides a solid background on the subject and highlights the research gap. However, the paper has only 11 references. It could be enhanced by including more recent references to studies on thin plate behavior under similar conditions.

2. It is hard to read the characters in Figure 1.

3. Table 1 provides essential material properties but could also include a brief explanation of how these values were chosen or sourced.

4. The study investigates both movable and immovable boundary conditions effectively. However, a more detailed explanation of how the movable and immovable conditions are implemented in the FEA would add clarity.

5. The development of an empirical formula for the SSSS-M boundary condition is a significant contribution. However, the formula's limitations should be discussed, including the range of validity and potential deviations for different materials or loading conditions.

6. The study could benefit from a more detailed discussion of the practical implications of these findings for engineering design.

Author Response

Reviewer # 2: Comments and Suggestions for the Authors

The authors would like to thank the reviewer for his constructive comments.

Our reply to reviewer's comments, are highlighted in yellow

1 The introduction provides a solid background on the subject and highlights the research gap. However, the paper has only 11 references. It could be enhanced by including more recent references to studies on thin plate behavior under similar conditions.

Additional references have been added to the reference list with explanatory text in the Introduction, although the topic of the present study is found to be less addressed by the various scholars.

  1. It is hard to read the characters in Figure 1.

Both Figure 1 and Figure 2 have been improved.

  1. Table 1 provides essential material properties but could also include a brief explanation of how these values were chosen or sourced.

The material values presented in Table 1 relates to Polycarbonate , as pointed up in the revised manuscript. This is a material used to manufacture multiwall rectangular plates as presented in the paper.

  1. The study investigates both movable and immovable boundary conditions effectively. However, a more detailed explanation of how the movable and immovable conditions are implemented in the FEA would add clarity.

Appendix A provides the necessary information to answer this comment.

  1. The development of an empirical formula for the SSSS-M boundary condition is a significant contribution. However, the formula's limitations should be discussed, including the range of validity and potential deviations for different materials or loading conditions.

The range of validity of expression (2) is now discussed in the revised manuscript. The investigation of the influence of material properties and various loading on the formula constants seems to be beyond the scope of this article and therefore not address in the present study.

  1. The study could benefit from a more detailed discussion of the practical implications of these findings for engineering design.

The practical implications of the present study are discussed and presented in the last paragraph of the Discussion section of the manuscript.

Reviewer 3 Report

Comments and Suggestions for Authors

Title: Behavior of Long Thin Rectangular Plates Under Normal Pressure - A Thorough Investigation

This study focuses on the long plate with a high aspect ratio in detail. The results reveal that long thin plates with movable boundary conditions show greater mid-width displacement compared to infinitely long plates, a phenomenon observed under both small and large deflection scenarios and across different boundary conditions. However, the following comments should be addressed before this paper can be considered for publication.

1. As a research paper, the authors should perform a review of the topic from the existing studies. The reference list only includes 11 papers indicating a more comprehensive review should be conducted.

2. The authors listed the material parameters in Table 1. However, the authors should provide more details about the material analyzed in this study. For instance, what type of material is used in the study? Also, the corresponding reference related to these parameters should be provided.

3. Since the finite element simulation is critical in this study. More details should be provided related to finite element simulation. For instance, the authors should provide details about large deformation handling, mesh control, element selection, etc. The corresponding justification behind the parameter selection for finite element simulation should also be provided.

4. In Figure 7, the authors utilized both Femap and Ansys for model analysis. It is important to provide the rationale behind selecting two different software platforms for this analysis. Also, the results from each software do not appear to align. This raises questions about the consistency and reliability of the results. The authors should provide a detailed explanation of how they validated the models in both software packages to ensure the accuracy and correctness of the output results.

5. The authors should include a separate conclusion section at the end of the study to succinctly summarize the key findings, implications, and contributions of their research. This section should point out the results and their significance in the field. Also, it should outline potential directions for future research.

Author Response

Reviewer # 3: Comments and Suggestions for the Authors

The authors would like to thank the reviewer for his constructive comments.

Our reply to reviewer's comments, are highlighted in yellow

This study focuses on the long plate with a high aspect ratio in detail. The results reveal that long thin plates with movable boundary conditions show greater mid-width displacement compared to infinitely long plates, a phenomenon observed under both small and large deflection scenarios and across different boundary conditions. However, the following comments should be addressed before this paper can be considered for publication.

  1. As a research paper, the authors should perform a review of the topic from the existing studies. The reference list only includes 11 papers indicating a more comprehensive review should be conducted.

The topic of the present study is limited in its available sources published in the literature. However additional references were added to provide the reader with more information regarding the present study.

  1. The authors listed the material parameters in Table 1. However, the authors should provide more details about the material analyzed in this study. For instance, what type of material is used in the study? Also, the corresponding reference related to these parameters should be provided.

Information about the material in Table 1 has been added in the revised manuscript

  1. Since the finite element simulation is critical in this study. More details should be provided related to finite element simulation. For instance, the authors should provide details about large deformation handling, mesh control, element selection, etc. The corresponding justification behind the parameter selection for finite element simulation should also be provided.

Appendix A was added to the manuscript, describing in detail the whole FEA process and its accuracy.

  1. In Figure 7, the authors utilized both Femap and Ansys for model analysis. It is important to provide the rationale behind selecting two different software platforms for this analysis. Also, the results from each software do not appear to align. This raises questions about the consistency and reliability of the results. The authors should provide a detailed explanation of how they validated the models in both software packages to ensure the accuracy and correctness of the output results.

The reason for using two different software was only to verify that the results were not influenced by an incidental bug in one of the systems. The differences between the two software were caused by the different number of elements. The results predicted by the Ansys code were based on coarse model stemming from the student version of the code available to the authors. Both Femap and Ansys have a very good non-linear capability which was required in the present study. As for validations, the authors could not find in the literature any empirical real test results for high AR plates. Also, theoretical models do not specifically address the high AR case. Therefore, FEA was used as a substitute to real tests and reliable theory, without additional validation. The added Appendix A describes the FEA in detail, including the convergence control.

  1. The authors should include a separate conclusion section at the end of the study to succinctly summarize the key findings, implications, and contributions of their research. This section should point out the results and their significance in the field. Also, it should outline potential directions for future research.

The Discussion and the Conclusions sections are now separated in the revised manuscript. A future research suggestion has been added to the discussion section. The authors believe that this paper will enable further research of the discovered phenomenon.

Round 2

Reviewer 1 Report

Comments and Suggestions for Authors